# N460S in PB2 and I163T in nucleoprotein synergistically enhance the viral replication and pathogenicity of influenza B virus

Yang Wang[1,2,3☯*], Yu Gao[1☯], Tianxin Ma[1☯], Yuting Ye[1], Chenyang Cao[1], Binqian Zou[1], Sulan Ye[1], Qingsheng Huang[1], Chunguang Yang[1], Shengfeng Li[1], Lixi Liang[1], Hongxuan Zhou[1], Zhiqi Zeng[1,3], Zifeng Yang[1,2,3,4*], Weiqi Pan[1,3*]

**1** State Key Laboratory of Respiratory Disease, National Clinical Research Center for Respiratory Disease, Guangzhou Institute of Respiratory Health, The First Affiliated Hospital of Guangzhou Medical University, Guangzhou, Guangdong, China, **2** Guangzhou National Laboratory, Guangzhou, Guangdong, China, **3** Respiratory Disease AI Laboratory on Epidemic and Medical Big Data Instrument Applications, Faculty of Innovation Engineering, Macau University of Science and Technology, Macau, China, **4** State Key Laboratory of Quality Research in Chinese Medicine, Macau University of Science and Technology, Macau, China

☯ These authors contributed equally to this work.
* panweiqi@gird.cn (WP); jeffyah@163.com (ZY); wang_yang@gzlab.ac.cn (YW)

## Abstract

Influenza B viruses (IBVs), though often overshadowed by influenza A viruses (IAVs), remain a significant global public health concern, particularly during seasons when they predominate. However, the molecular mechanisms underlying IBV pathogenicity remain largely unknown. In this study, we identified two amino acid substitutions, PB2-N460S and NP-I163T, from IBV clinical isolates with distinct replication and pathogenicity profiles. Using reverse genetics, we generated recombinant IBV viruses to evaluate the impact of these substitutions. *In vitro* and *in vivo* infections revealed that viral replication and pathogenicity were not significantly affected by either substitution alone but were substantially enhanced when both substitutions occurred together. Lung transcriptomics in mice infected with virus containing PB2-N460S/NP-I163T substitutions showed heightened immune activation. This was characterized by upregulated transcription of antiviral and immune-related genes, contributing to excessive inflammation and severe disease outcomes. Mechanistic investigations demonstrated that each substitution independently increased protein expression and strengthened PB2-NP interactions. However, only the combined presence of PB2-N460S and NP-I163T significantly enhanced polymerase activity. Structural modeling indicated that PB2–460 residue is positioned at the PB2-NP interface, while NP-163 site resides distally, suggesting an indirect functional interplay. These findings provide new insights into the molecular determinants of IBV pathogenesis, highlighting the synergistic effect of PB2-N460S and NP-I163T in enhancing viral fitness and worsening disease outcomes.

**Data availability statement:** All next-generation sequencing data generated in this study have been deposited in the Gene Expression Omnibus (GEO) under Accession No. GSE289500 and in the NCBI Sequence Read Archive (SRA) under BioProject No. PRJNA1294812. These datasets are publicly accessible.

**Funding:** This work was supported by the National Key Research and Development Program of China (Grant No. 2024YFE0214800 to W.P.), the National Natural Science Foundation of China (Grant No. 82361168672 to Z.Y.), the Science and Technology Development Fund of Macau SAR (Grant Nos. FDCT 0111/2023/AFJ to Z.Y. and 005/2022/ALC to Z.Z.), and the State Key Laboratory of Respiratory Disease (Grant No. SKLRD-Z-202405 to T.M.). The funders had no role in study design, data collection and analysis, decision to publish, or preparation of the manuscript.

**Competing interests:** The authors have declared that no competing interests exist.

## Author summary

Influenza B virus (IBV) remains a significant public health concern, contributing to a substantial portion of seasonal influenza cases and sometimes emerging as the dominant circulating strain. Despite its clinical impact, the molecular mechanisms underlying IBV pathogenicity are not well understood. In this study, we identified two amino acid substitutions, N460S in PB2 and I163T in NP, from IBV clinical isolates exhibiting distinct replication and pathogenicity phenotypes. Using reverse genetics, we demonstrated that although neither mutation alone significantly affected viral replication or pathogenicity, their combination synergistically enhanced both traits. Our findings provide critical insights into the functional interplay between PB2 and NP, two essential components of the viral ribonucleoprotein (RNP) complex. We found that these substitutions increase protein expression and strengthen PB2-NP interaction, yet only in combination do they enhance polymerase activity, suggesting a cooperative role in viral RNA synthesis. Structural modeling further revealed that PB2-460 is located at the PB2-NP interface, while NP-163 is not, implying an indirect regulatory mechanism. These results expand our understanding of how PB2 and NP functionally interact to modulate IBV replication and virulence, shedding light on potential molecular determinants that drive IBV pathogenesis.

## Introduction

Seasonal influenza remains a significant global public health challenge, affecting up to 1 billion people annually and leading to 3–5 million severe cases and approximately 290,000–650,000 deaths worldwide [1–3]. While influenza A viruses (IAVs) are typically responsible for the majority of cases, influenza B viruses (IBVs) contribute to approximately 25% of annual infections, with this proportion varying greatly across influenza seasons [4,5]. In certain years, IBVs can even become the predominant circulating virus, accounting for up to 80% of all influenza cases [6–9]. Moreover, IBV infections can lead to a spectrum of clinical outcomes, ranging from mild symptoms, such as fever, cough, and body aches, to severe complications, including pneumonia, acute respiratory distress syndrome, and even mortality [10]. Research has found no statistically significant difference in disease severity or complications between pediatric patients infected with IAVs and those with IBVs [11,12]. This underscores the need for a deeper understanding of the pathogenesis of IBVs to improve prevention and treatment strategies.

IBVs share a similar genomic structure with IAVs, consisting of eight linear negative-sense, single-stranded RNA segments that encode at least 11 viral proteins [13]. These proteins include the heterotrimeric polymerase (PB2, PB1 and PA subunits), nucleoprotein (NP), surface glycoproteins (HA, NA and NB), matrix protein (M1), ion channel (M2), non-structural protein (NS1), and nuclear export protein (NEP) [10]. In both virus types, each RNA segment is packaged into a

ribonucleoprotein (RNP) complex, in which the 5′ and 3′ terminal of viral RNA are bound by one polymerase heterotrimer while the RNA is covered by multiple NP monomers [14]. Despite these similarities, IAV and IBV RNPs differ markedly in sequence, structure, and function [15]. Cryo-EM tomography shows that IAV RNPs adopt a tightly ordered "7 + 1" oval arrangement [16], whereas IBV RNPs appear as loosely packed, variably oriented bundles without a defined "7 + 1" pattern [17]. At the polymerase level, IAV and IBV polymerase heterotrimers engage host RNA polymerase II and ANP32 family cofactors via distinct interfaces and suggesting distinct RNA-synthesis and RNP-assembly mechanisms [18,19]. Sequence differences in the PB2 cap-binding domain also give IBV PB2 an inverted substrate-recognition mode and reduced m⁷GDP affinity compared to IAV PB2 [20]. Finally, IBV NP carries an a unique N-terminal extension of approximately 50–70 residues—essential for viral viability, nuclear import, and efficient transcription and replication—that is absent in IAV NP [21,22].

The mammalian pathogenicity of influenza viruses is predominantly determined by specific amino acid substitutions. Extensive research has identified key genetic determinants that contribute to the replication and pathogenicity of IAVs. Notable examples include the PB2-E627K [23,24] and PB2-D701N [25] substitutions, as well as the NP-Q357K [26] substitution. Furthermore, a previous study revealed that NP-D88 directly engages PB2 to orchestrated viral replication [27]. Notably, PB2-D701N and NP-N319K substitution synergistically augmented IAV replication and pathogenicity by enhancing importin α1 binding affinity, thereby facilitating viral ribonucleoprotein nuclear import [28,29]. In contrast, the molecular drivers of IBV pathogenicity remain poorly characterized. Limited studies have identified putative virulence markers, such as PA-K338R [30], PB2-F406Y/W359F [31], M1-N221S [32], and NA-N342D [33], though mechanistic insights into their functional contributions are lacking.

In our previous study, we isolated the parental B/Guangzhou/0215/2012 (GZ0215) viral isolate from a patient in Guangzhou, China, and characterized it in ferret, tree shrew, and mouse models. That work found mild symptoms in ferrets and tree shrews but lethality in mice [34]. Building on these findings, we have now plaque-purified two clonal variants (B/Guangzhou/0215-01/2012 and B/Guangzhou/0215-06/2012) from that same clinical sample. These two isolates exhibited significant differences in viral replication and virulence *in vitro* and *in vivo* viral challenge assays. The whole genome analysis identified two nonsynonymous substitutions: PB2-N460S and NP-I163T. To delineate their functional contributions, we utilized reverse genetics to generate recombinant IBV strains harboring individual or combined substitutions. Our findings revealed the synergistic impact of PB2-N460S and NP-I163T on viral polymerase activity, replication efficiency, and pathogenicity. This study offers new insights into the molecular determinants of IBV pathogenicity, the potential interactions between PB2 and NP proteins, and advances our understanding of IBV biology, ultimately supporting public health surveillance efforts.

## Results

### Two distinct clonal strains of an IBV clinical isolate exhibit different virulence phenotypes

The influenza B virus strain B/Guangzhou/0215/2012 (GZ0215) was recovered from the oropharyngeal swab of a 12-year-old male outpatient who presented on February 16, 2012 with acute upper respiratory tract infection (fever 39.3 °C, nasal congestion, rhinorrhea, sneezing) and received no antiviral treatment. Whole genomic sequencing of this stock identified amino acid polymorphism in PB2, PA, and NP proteins (Table 1).

To resolve viral heterogeneity, we performed three rounds of plaque purification, yielding two isolates, referred to as B/Guangzhou/0215-01/2012 (GZ0215-01) and B/Guangzhou/0215-06/2012 (GZ0215-06). These isolates exhibited distinct plaque morphologies (Fig 1A). The average plaques size produced by GZ0215-06 was 3.9-fold larger than that of GZ0215-01 ($p<0.0001$) (Fig 1B). To assess the association between viral replication in cell culture and pathogenicity in mammalian models, BALB/c mice were intranasally challenged with tenfold serial dilutions of each strain: $10^2$–$10^5$ TCID$_{50}$ for GZ0215-01 and $10^1$–$10^4$ TCID$_{50}$ for GZ0215-06. GZ0215-06 demonstrated marked pathogenicity (median lethal dose

**Table 1. Analysis of genomic polymorphisms in B/Guangzhou/0215/2012 (GZ0215) isolated from an oropharyngeal sample collected in Guangzhou, China, 2012.**

| Segment/Protein | Nucleotide allele | | Amino acid allele | | Coverage | Percentile |
|---|---|---|---|---|---|---|
| | Position[a] | Codons | Position[b] | Residues | | |
| PB2 | 781-783 | ATG/GTG | 261 | M/V | 11186 | 82.67/16.96 |
| | 1378-1380 | AGT/AAT | 460 | S/N | 14276 | 81.84/17.91 |
| PA | 877-879 | AAT/AGT | 293 | N/S | 42053 | 95.34/4.26 |
| | 1012-1014 | AAG/AGG | 338 | K/R | 47592 | 94.45/5.00 |
| | 1483-1485 | ATG/ATA | 495 | M/I | 49163 | 92.37/7.36 |
| NP | 487-489 | ACC/ATC | 163 | T/I | 19179 | 94.72/4.82 |

[a]The nucleotide positions in the viral genomes were annotated starting from the first codon of the coding region; [b] the amino acid positions were annotated based on the translated open reading frames.

$[LD_{50}] = 10^{2.5}$ $TCID_{50}$), whereas GZ0215-01 did not cause noticeable body weight loss in infected mice, even at the highest challenge dose of $10^5$ $TCID_{50}$ (Fig 1C and 1D). Whole-genome sequencing confirmed genomic identity between isolates except for two nonsynonymous substitutions: PB2-460N/NP-163I in GZ0215-01 versus PB2-460S/NP-163T in GZ0215-06 (Fig 1E).

### PB2-N460S/NP-I163T substitutions synergistically enhance viral replication of IBVs in cells

To investigate the individual and combined effects of PB2-N460S and NP-I163T substitutions on viral replication and pathogenicity, we employed reverse genetics to generate four recombinant IBV viruses in the GZ0215-01 genetic background. These recombinant viruses include: rgPB2$_{460N}$/NP$_{163I}$, which is identical to the wild-type GZ0215-01, rgPB2$_{460S}$/NP$_{163I}$ (PB2-N460S single substitution), rgPB2$_{460N}$/NP$_{163T}$ (NP-I163T single substitution), and rgPB2$_{460S}$/NP$_{163T}$ (double substitutions, wild-type GZ0215-06).

Plaque morphology analysis in MDCK cells revealed no significant differences among rgPB2$_{460N}$/NP$_{163I}$, rgPB2$_{460S}$/NP$_{163I}$, and rgPB2$_{460N}$/NP$_{163T}$ ($p > 0.05$ for all pairwise comparisons). However, the rgPB2$_{460S}$/NP$_{163T}$ with double substitutions exhibited markedly larger plaque sizes compared to the other three isogenic viruses ($p < 0.0001$; Fig 2A and 2B). Viral growth kinetics analysis further supported these findings, with rgPB2$_{460S}$/NP$_{163T}$ exhibiting significantly enhanced replication capacity across all time points ($p = 0.0226$ to $p < 0.0001$ versus other variants at each timepoint; Fig 2C). Furthermore, we ruled out differences in viral entry by performing viral attachment and internalization assays in MDCK cells (S2 Fig). To detect potential host-adapted mutations, we sequenced viral genomes from MDCK cell supernatants at 72 hpi for each recombinant virus. Across all four viruses, we detected only low-frequency (<15%) host-adaptive substitutions at a few positions in the viral genome outside PB2–460 and NP-163, indicating the absence of high-frequency adaptive mutations during *in vitro* passage. These results suggest that the concurrent presence of PB2-N460S and NP-I163T substitutions is required to enhance IBV replication, as neither substitution alone significantly affects replication.

### PB2-N460S/NP-I163T substitutions synergistically enhance pathogenicity of IBVs in mice

To further assess the individual and combined impact of PB2-N460S and NP-I163T substitutions on viral pathogenicity *in vivo*, we intranasally infected BALB/c mice with serial doses of the four above-mentioned recombinant IBVs (Figs 3A-3C and S1). Notably, the rgPB2$_{460S}$/NP$_{163T}$ with double substitutions demonstrated marked lethality with an $LD_{50}$ of $10^{2.5}$ $TCID_{50}$, whereas mice infected with rgPB2$_{460N}$/NP$_{163I}$ (wild-type GZ0215-01), rgPB2$_{460S}$/NPI$_{163I}$ (PB2 single substitution), or rgPB2$_{460N}$/NP$_{163T}$ (NP single substitution) survived through the 14-day observation period even at the maximum challenge

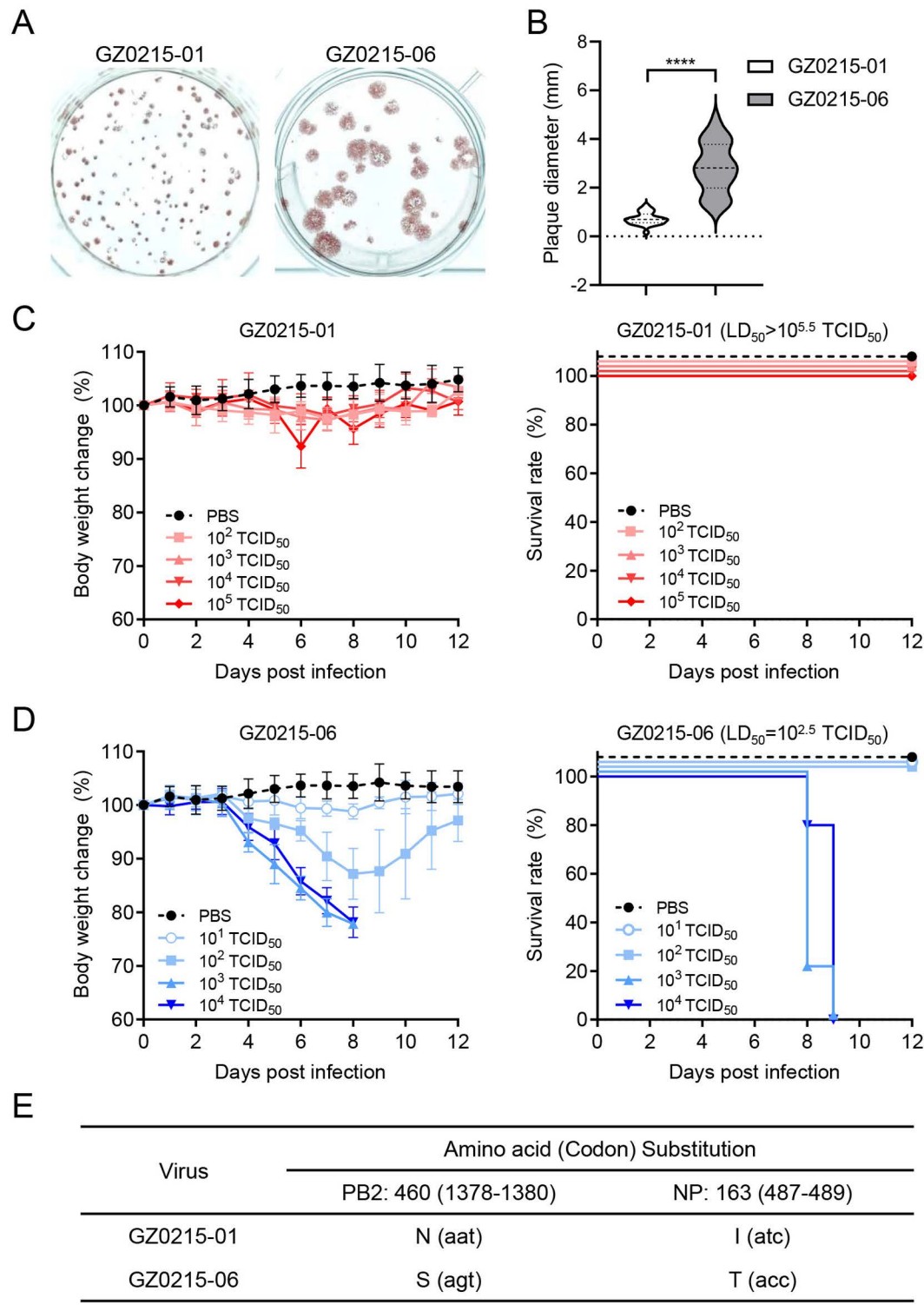

**Fig 1. Two distinct clonal strains of an IBV clinical isolate exhibit different virulence phenotypes.** (A) A pair of IBVs B/Guangzhou/0215-01/2012 (GZ0215-01) and B/Guangzhou/0215-06/2012 (GZ0215-06) were isolated from the oropharyngeal swab of a single patient using plaque clonal purification. Plaque morphologies were developed at 3 dpi on MDCK cells and visualized by immunostaining. (B) Plaque diameters for each IBV were measured using Adobe Photoshop (CC 2019). (C and D) Body weight changes (left panel) and survival rates (right panel) were monitored in six-week-old female

BALB/c mice intranasally infected with serial doses of GZ0215-01 (C), GZ0215-06 (D), or PBS as a control. Five animals were used per group. The median lethal doses ($LD_{50}$) of the two isolates were calculated based on the survival rates from panels C and D. (E) Whole viral genome analysis via Sanger sequencing revealed only two amino acid substitutions between these isolates: N460S in the PB2 protein and I163T in the NP protein, resulted from codon changes aat1378-1380agt and atc487-489acc, respectively. ****, $p < 0.0001$.

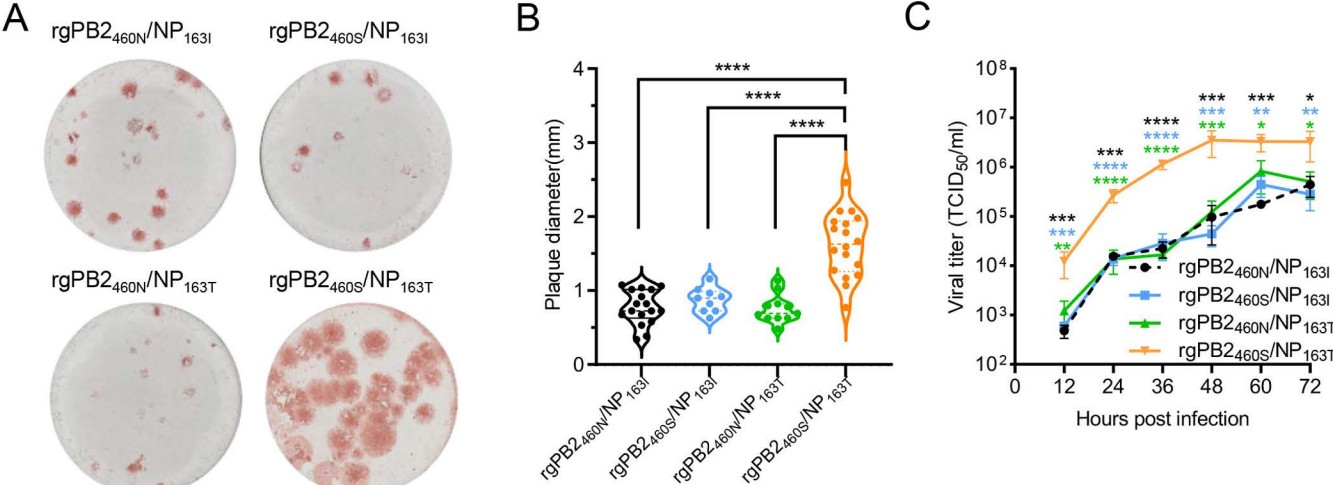

**Fig 2. PB2-N460S/NP-I163T substitutions synergistically enhance viral replication of IBVs in cells.** (A) Four recombinant IBVs were constructed through reverse genetics, each derived from the whole genome of GZ0215-01. These recombinant viruses included: one without any amino acid substitution (denoted as rgPB2$_{460N}$/NP$_{163I}$), one with the PB2-N460S substitution (denoted as rgPB2$_{460S}$/NP$_{163I}$), another with the NP-I163T substitution (denoted as rgPB2$_{460N}$/NP$_{163T}$), and a fourth with both PB2-N460S and NP-I163T substitutions (denoted as rgPB2$_{460S}$/NP$_{163T}$). Plaque morphologies were developed at 3 days post-infection (dpi) on MDCK cells and were visualized by immunostaining. (B) Plaque diameters for each recombinant IBV were determined by Adobe Photoshop (CC 2019). (C) Growth kinetics of the indicated recombinant viruses in MDCK cells at an MOI of 0.01. Statistical significance is indicated using asterisks in different colors for clarity: black, blue, and green asterisks denote significant differences of rgPB2$_{460N}$/NP$_{163I}$ vs. rgPB2$_{460S}$/NP$_{163T}$, rgPB2$_{460S}$/NP$_{163I}$ vs. rgPB2$_{460S}$/NP$_{163T}$, and rgPB2$_{460N}$/NP$_{163T}$ vs. rgPB2$_{460S}$/NP$_{163T,}$ respectively. *, $p < 0.05$; **, $p < 0.01$; ***, $p < 0.001$; ****, $p < 0.0001$.

dose of $10^5$ TCID$_{50}$. These results indicate that concurrent presence of both PB2-N460S and NP-I163T substitutions are strictly required for lethal pathogenicity in murine model.

To assess the impact of PB2-N460S and NP-I163T substitutions on viral replication in mouse airway, we measured viral load in the nasal turbinate and lungs of mice infected with $10^4$ TCID$_{50}$ of each recombinant virus. All test viruses replicated in the respiratory tracts of mice, peaking at 3 dpi both in the turbinate and lungs (Fig 3D and 3E). The viral loads in the mice infected with rgPB2$_{460S}$/NP$_{163T}$ were significantly higher than in those infected with all other three recombinant viruses, both in the nasal turbinate ($p = 0.0013 \sim p < 0.0001$) (Fig 3D) and lungs ($p = 0.0023 \sim p < 0.0001$) (Fig 3E) throughout the infection course. No notable differences were observed between mice infected with rgPB2$_{460N}$/NP$_{163I}$, rgPB2$_{460S}$/NP$_{163I}$, or rgPB2$_{460N}$/NP$_{163T}$ in either turbinate or lungs.

To explore whether PB2-N460S and/or NP-I163T substitutions enhance the inflammatory response and pathology in mice, we determined gene expression levels of type I IFN (i.e., IFN-α), and representative pro-inflammatory cytokines and chemokines (i.e., IL-1β, IL-6, IL-8, TNF, TNF-α, CCL-5/RANTES and IL-10) (S1 Table) in the mouse lung tissues collected at 1, 3, and 5 dpi. Compared to the control mice inoculated with PBS, all four recombinant viruses stimulated inflammatory cytokines and chemokines responses in lung tissues of infected mice, particularly at 5 dpi (Fig 3F). Mice infected with rgPB2$_{460S}$/NP$_{163T}$ had significantly elevated cytokine/chemokine responses in the lungs than those infected with other viruses. This hyperinflammatory state correlated with enhanced histopathological manifestations in lung tissues, as evidenced by H&E staining showing exacerbated pulmonary pathology at 5 dpi (Fig 3G). Compared to wild-type and single

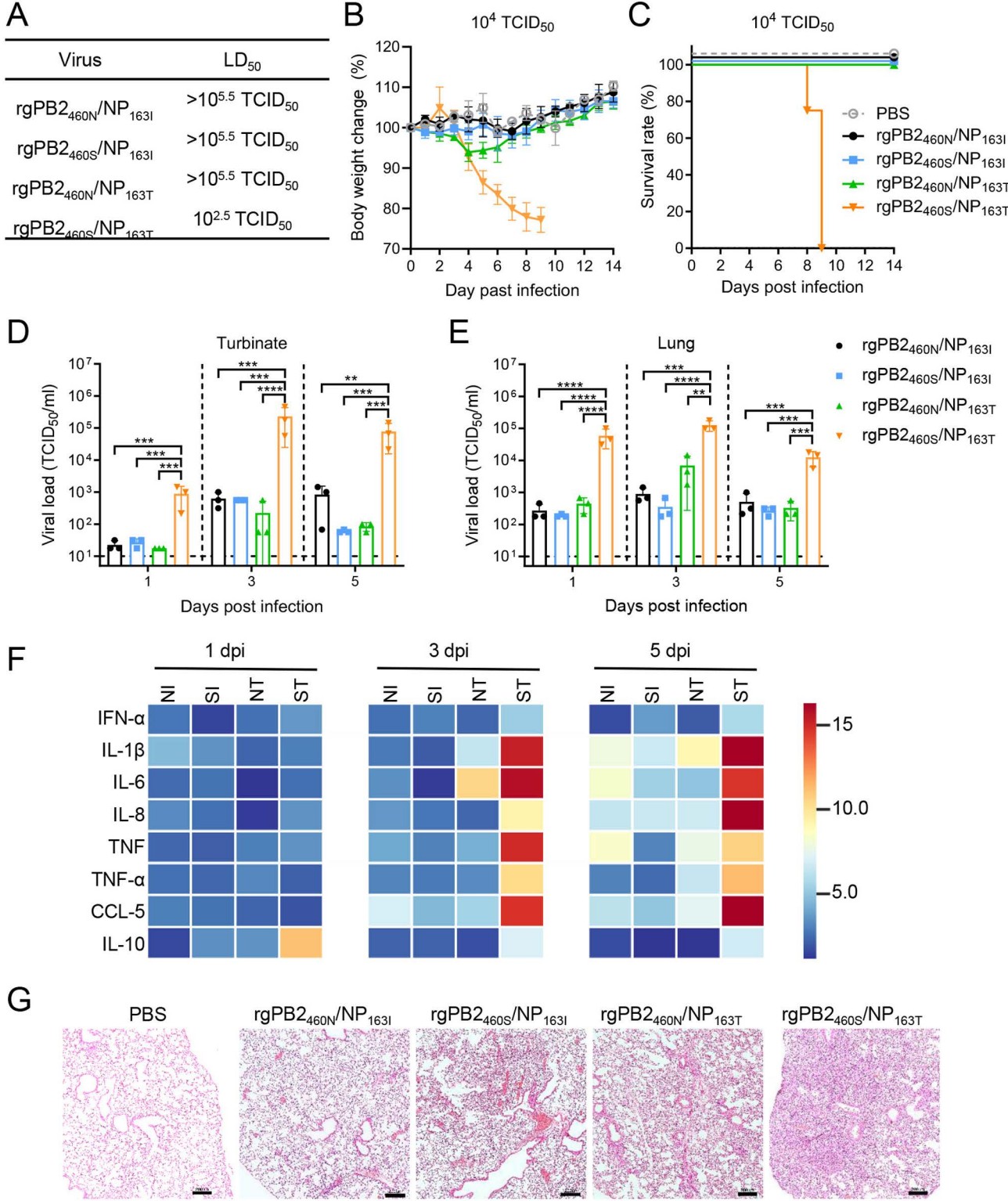

**Fig 3. PB2-N460S/NP-I163T substitutions synergistically enhance pathogenicity of IBVs in mice.** (A) LD$_{50}$ of the indicated recombinant IBVs. Body weight changes (B) and survival rates (C) of six-week-old female BALB/c mice intranasally infected with $10^4$ TCID$_{50}$ of each recombinant IBV or

PBS. Four animals were used in each group. Virus titers in nasal turbinate (D) and lungs (E) of mice infected with $10^4$ PFU of the indicated viruses. Circles, squares or triangles represent viral titers from individual mice (n = 3 for each dpi). The limit of detection is indicated by the horizontal dashed line. **, $p < 0.01$; ***, $p < 0.001$; ****, $p < 0.0001$. (F) Heatmap of average relative mRNA expression of cytokines and chemokines in lung tissues from mice infected with $10^4$ $TCID_{50}$ of $rgPB2_{460N}/NP_{163I}$ (NI), $rgPB2_{460S}/NP_{163I}$ (SI), $rgPB2_{460N}/NP_{163T}$ (NT), or $rgPB2_{460S}/NP_{163T}$ (ST) viruses at indicated dpi. The heatmap was generated with HemI 2.0 software. (G) Hemotoxylin and eosin (H&E) staining of lung sections from mice inoculated with $10^4$ $TCID_{50}$ of the indicated virus or PBS at 5 dpi. Magnification, 100×; scale bar, 200 μm.

substitution viruses, the lung tissues of mice infected with $rgPB2_{460S}/NP_{163T}$ show increased inflammatory cell infiltration, more pronounced disruption of alveolar architecture, and greater cellular shedding in the bronchial lumen.

To assess potential host-adaptive mutations, we sequenced viral genomes from mouse lung tissues harvested at 5 dpi for each recombinant virus. No adaptive mutations were detected at PB2-460 or NP-163 in any sample, and no high-frequency (>50%) mutations were identified. Although a PB2-527 polymorphism was present at moderate frequency (25–49%) across all samples, its prevalence in the PB2-460S/NP-163T double mutant was lower than in the other variants. These data confirm that host-adaptive substitutions did not drive the enhanced replication or pathogenicity of the $rgPB2_{460S}/NP_{163T}$ virus.

To test whether the synergistic effect of PB2-N460S/NP-I163T extends beyond GZ0215, we introduced the reverse mutations (PB2-S460N and NP-T163I) into a more recently circulating IBV strain, B/Guangzhou/50/2022 (Victoria lineage, clade V1A.3a.1; GISAID isolate EPI_ISL_19888228), which we confirmed is lethal in mice. Using reverse genetics, we rescued the parental GZ50 (PB2-460S/NP-163T; "GZ50-ST"), the single-substitution mutants GZ50-NT (PB2-460N/NP-163T) and GZ50-SI (PB2-460S/NP-163I), and the double-reverse mutant GZ50-NI (PB2-460N/NP-163I). In MDCK cells, GZ50-NI, GZ50-SI, and GZ50-NT produced indistinguishable plaque sizes and comparable replication kinetics, whereas GZ50-ST generated significantly larger plaques and higher titers (S2A–S2C Fig). In BALB/c mice challenged with $10^4$ $TCID_{50}$ of each virus, mice infected with GZ50-NI, GZ50-SI, or GZ50-NT exhibited mild to no body-weight loss, but all GZ50-ST–infected mice lost > 25% body weight and were humanely euthanized (S2D–S2E Fig). Lung titers at 1, 3, and 5 dpi were highest in GZ50-ST–infected mice compared to the three mutants (S2F Fig). These results indicate that, in the GZ50 background, restoring PB2-460N/NP-163I attenuates replication and pathogenicity, mirroring our GZ0215 findings and confirming that the PB2-N460S/NP-I163T synergy is conserved across diverse IBV strains.

Taken together, our results indicate that the PB2-N460S and NP-I163T substitutions, when present together, synergistically enhance both the replication and pathogenicity of IBVs in mice.

**PB2-N460S/NP-I163T substitutions up-regulate innate immune responses**

To investigate the effects of PB2-N460S and NP-I163T double substitutions on host immune and antiviral response, we analyzed the transcriptional profiles of mouse lung tissues harvested at 5 dpi (Fig 4). In total, we identified 1,092 differentially expressed genes (DEGs) in mice infected with $rgPB2_{460N}/NP_{163I}$ (894 up-regulated and 198 down-regulated) and 2,674 DEGs in mice infected with $rgPB2_{460S}/NP_{163T}$ (1,559 up-regulated and 1,115 down-regulated), with an adjusted $p$-value < 0.01 and log2 fold-change ≥1. Among the DEGs, 791 genes were commonly up-regulated in both groups, while 103 genes were uniquely up-regulated in the $rgPB2_{460N}/NP_{163I}$ group, and 768 genes were uniquely up-regulated in the $rgPB2_{460S}/NP_{163T}$ group (Fig 4A).

Notably, we observed significant up-regulation of a variety of genes related to cytokine/chemokine signaling pathway (e.g., Il6, Ccl5, Tnf), interferon responses (e.g., Ifnb1, Ifng, Isg15), antiviral defense (e.g., Rsad2, Zbp1, Oas3, Trim family), and immune-related functions (e.g., Cd80, Tlr3) in the $rgPB2_{460S}/NP_{163T}$ infected mice (Fig 4B). Pathway enrichment analysis revealed significant activation of innate immunity-related processes in the $rgPB2_{460S}/NP_{163T}$ group compared to the $rgPB2_{460N}/NP_{163I}$ group (Fig 4C-4D). These processes were prominently associated with IL-1 biosynthesis, cytokine-mediated signaling cascades, and innate immune response regulation. Specifically, immune-related genes

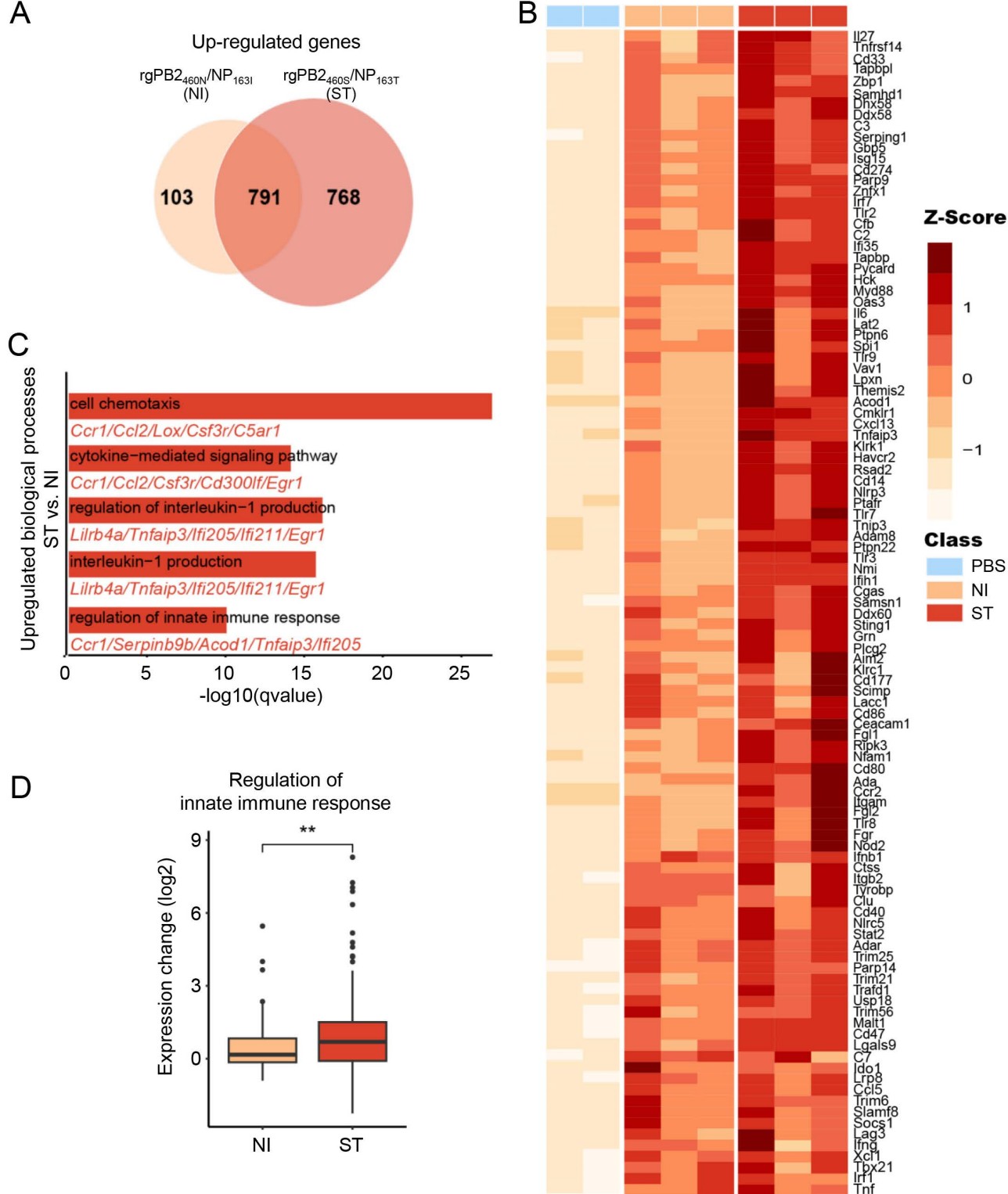

**Fig 4. PB2-N460S/NP-I163T substitutions up-regulate innate immune responses.** Six-week-old female BALB/c mice intranasally infected with $10^4$ TCID$_{50}$ of indicated IBVs or PBS as a control (n = 3 for infection groups and n = 2 for control group). Lungs were collected at 5 dpi, and total RNA

was isolated for RNA sequencing using the illumina NovaSeq 6000 platform. (A) Venn diagram shows the common and unique up-regulated genes in rgPB2$_{460N}$/NP$_{163I}$ and rgPB2$_{460S}$/NP$_{163T}$. (B) Heatmap illustrates the normalized expression of the most significantly differentially expressed genes between the two recombinant virus infection groups. (adjusted p-Value < 0.01, log2 fold-change ≥ 1). (C) Bar plot represents the differentially up-regulated biological processes between the rgPB2$_{460N}$/NP$_{163I}$ and rgPB2$_{460S}$/NP$_{163T}$ infection groups. (D) Boxplot displays the fold changes of genes involved in regulation of innate immune response between the rgPB2$_{460N}$/NP$_{163I}$ and rgPB2$_{460S}$/NP$_{163T}$ infection groups.

including Ccr1, Ccl2, and Serpinb9b exhibited pronounced enrichment in the rgPB2$_{460S}$/NP$_{163T}$ group, suggesting amplified inflammatory response signaling.

To assess whether these differences in innate immune response were linked to viral replication and virulence, we performed single-sample gene set enrichment analysis (ssGSEA) for the "Regulation of innate immune response" pathway and correlated the resulting enrichment scores with both respiratory tract viral titers and body-weight loss. Enrichment scores showed strong positive correlations with nasal turbinate titer (R = 0.92, $p$ = 0.0083), lung titer (R = 0.95, $p$ = 0.0043), and body-weight loss (R = 0.86, $p$ = 0.0274) at 5 dpi (S4 Fig). These results indicate that the enhanced innate immune responses in rgPB2$_{460S}$/NP$_{163T}$ infected mice are closely associated with increased viral replication and pathogenicity.

These findings indicate that the combinatorial PB2-N460S and NP-I163T substitutions hyperactivate innate immune signaling, triggering aberrant upregulation of interferon-stimulated genes (ISGs) and proinflammatory cytokines, which is largely associated with their enhanced viral replication.

## PB2-N460S/NP-I163T mutations synergistically augment polymerase activity through enhanced PB2 and NP protein expression and interaction

To elucidate the molecular mechanisms underlying the enhanced viral replication and pathogenicity conferred by PB2-N460S/NP-I163T substitutions, we systematically investigated their effects on protein expression and polymerase complex functionality. PB2-Myc (with 460N and 460S) and NP (with 163I and 163T) genes, derived from GZ0215-01, were cloned into mammalian expression vector pCAGGS. HEK293T cells transiently transfected with these plasmids, were lysed at 24 h post-transfection for parallel quantification of transcriptional and translational outputs via qRT-PCR and Western blot. Strikingly, PB2$_{460S}$ exhibited a 2.5-fold increase in mRNA level compared to PB2$_{460N}$ ($p$ = 0.0013, Fig 5A), with concordant upregulation of Myc-tagged protein expression (1.2-fold, $p$ = 0.0058, Fig 5B). Similarly, NP$_{163T}$ demonstrated improved mRNA level (29.7-fold, $p$ < 0.0001) and NP protein level (2.9-fold, $p$ = 0.0244) relative to NP$_{163I}$ (Fig 5C-5D), confirming mutation-dependent augmentation of viral protein biosynthesis.

Previous studies have demonstrated that interaction between PB2 and NP is critical for viral ribonucleoprotein (vRNP) assembly and regulation of viral RNA synthesis [35,36]. To further investigate the effect of these substitutions on PB2-NP interaction, we employed co-immunoprecipitation (Co-IP) assays in HEK293T cells co-transfected with bidirectional pM vectors encoding GZ0215-01 derived PB2-Myc (460N/S), NP (163I/T), PB1, and PA genes. Consistent with single-gene expression profiles (Fig 5B and 5D), PB2-N460S and NP-I163T substitutions increased the PB2 and NP expression levels, respectively, in the context of the complete vRNP complex (PB2: 1.7-fold, $p$ = 0.0005; NP: 2.2-fold, $p$ = 0.0086; Fig 5E and 5F). Crucially, Co-IP results revealed that the of both PB2-N460S and NP-I163T substitutions led to stronger PB2-NP interaction, as evidenced by the increased intensity of the PB2 band in IP samples (SI vs. NI: 5.0-fold, $p$ = 0.0367; NT vs. NI: 7.0-fold, $p$ = 0.0039). Furthermore, compared to wild-type GZ0215-01 and single substitutions, PB2-N460S and NP-I163T synergistically strengthened PB2-NP binding. This enhanced interaction was functionally validated through a *firefly* luciferase-based minigenome assay (pM backbone). The PB2$_{460S}$/NP$_{163T}$ with combinatorial substitutions showed 1.8~3.1 times higher polymerase activity compared to wild-type and single substitution groups ($p$ = 0.0049~0.0417, Fig 5G), with no significant differences observed between the wild-type and single substitution groups.

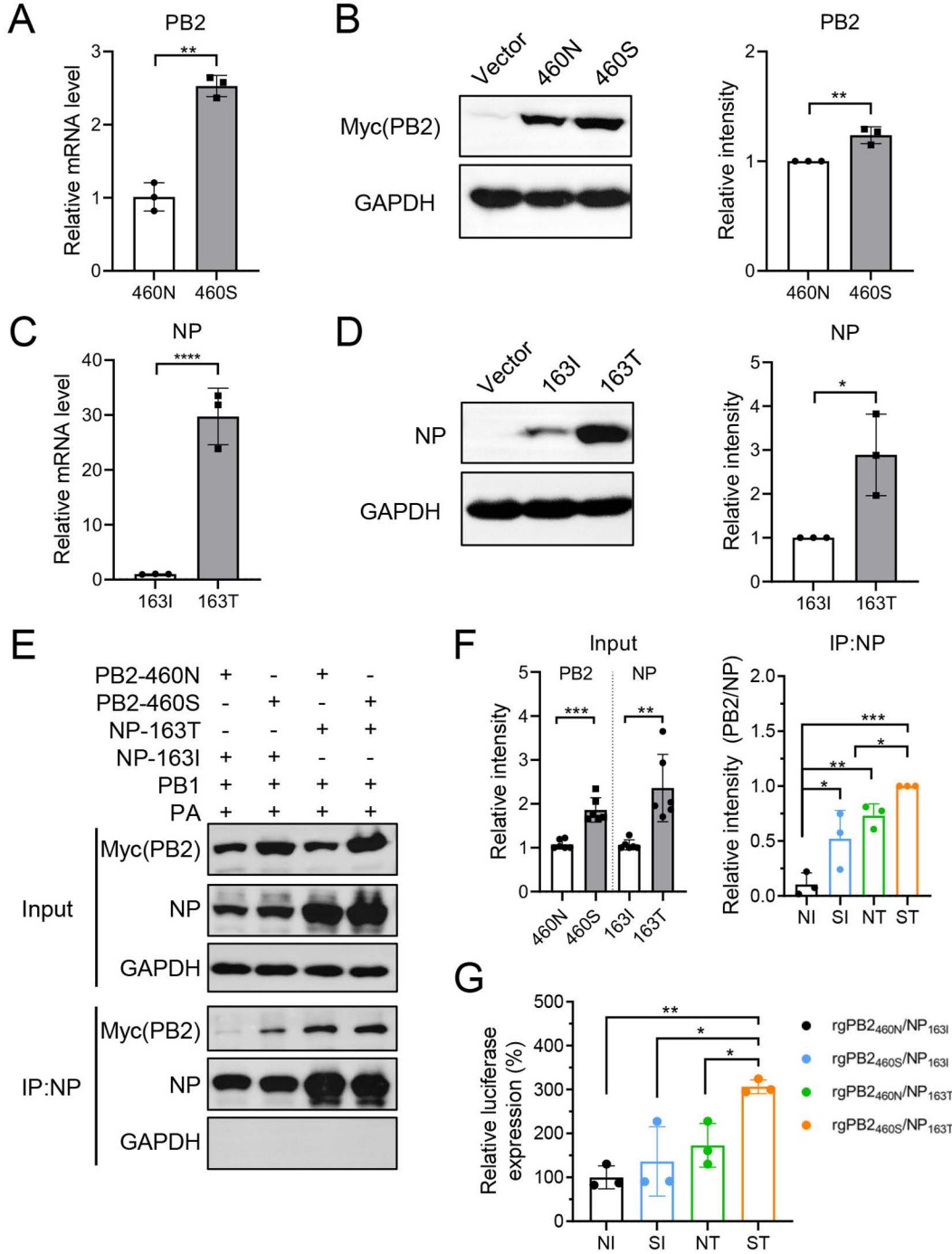

**Fig 5. PB2-N460S/NP-I163T mutations synergistically augment polymerase activity through enhanced PB2 and NP protein expression and interaction.** (A-D) HEK 293T cells were transfected with expression plasmids pCAGGS encoding GZ0215-01 derived PB2-Myc (with 460N or 460S) (A and B), or NP (with 163I or 163T) (C and D). As a control, empty pCAGGS vector was used. 24 hours post-transfection, cells were harvested, and relative mRNA levels of PB2 (A) and NP (C) were measured by qRT-PCR. hGAPDH was used as an internal reference. Protein expression levels of PB2-Myc (left panel of B) and NP (left panel of D) were determined by Western blot using rabbit anti-Myc tag polyclonal antibody and rabbit anti-NP polyclonal antibody, respectively. GAPDH was used as a loading control. The band intensities were quantified using ImageJ software (http://rsb.info.nih.gov/ij/) (right panels of B and D). (E-F) HEK 293T cells were transfected with bidirectional expression plasmids pM encoding GZ0215-01 derived PB2-Myc (with 460N or 460S) and NP (with 163I or 163T), along with PB1 and PA. Co-immunoprecipitation was performed using ProteinA/G magnetic beads

conjugated to NP antibody. Precipitated eluates and non-precipitated whole cell lysates were subjected to SDS-PAGE, and PB2 and NP were detected by Western blot (E). GAPDH was used as a loading control. The relative levels of input PB2 and NP proteins (normalized by GAPDH, left panel of F) and the relative levels of PB2 in IP samples (normalized by NP protein, right panel of F) were quantified using ImageJ software. (G) Polymerase activity. HEK 293T cells were co-transfected with PB2, PB1, PA, and NP plasmids from the indicated recombinant IBV mutants, along with a firefly luciferase reporter template flanked by the NP untranslated regions of IBV and a Renilla luciferase expression plasmid as an internal control. Dual-luciferase activity was measured 24 hours post-transfection. *, $p < 0.05$; **, $p < 0.01$; ***, $p < 0.001$; ****, $p < 0.0001$.

Given the unresolved structural architecture of IBV vRNP complexes, we employed AlphaFold 3 for *de novo* prediction of NP-polymerase docking conformations. Computational modeling revealed that PB2 residue 460 spatially colocalized within a putative NP-binding interface (Fig 6A). PRODIGY calculations then showed that the PB2-N460S substitution yielded a more favorable binding free energy (ΔG = −7.1 vs. −6.5 kcal/mol) and a lower dissociation constant ($Kd = 6.3 \times 10^{-6}$ M vs. $1.6 \times 10^{-5}$ M at 25 °C) (Fig 6B and 6C), indicating tighter PB2–NP binding. We speculate this stabilization arises from reduced steric hindrance due to the shortened side chain after the N-to-S substitution, enabling closer proximity between PB2 and NP-4 in the 460 region.

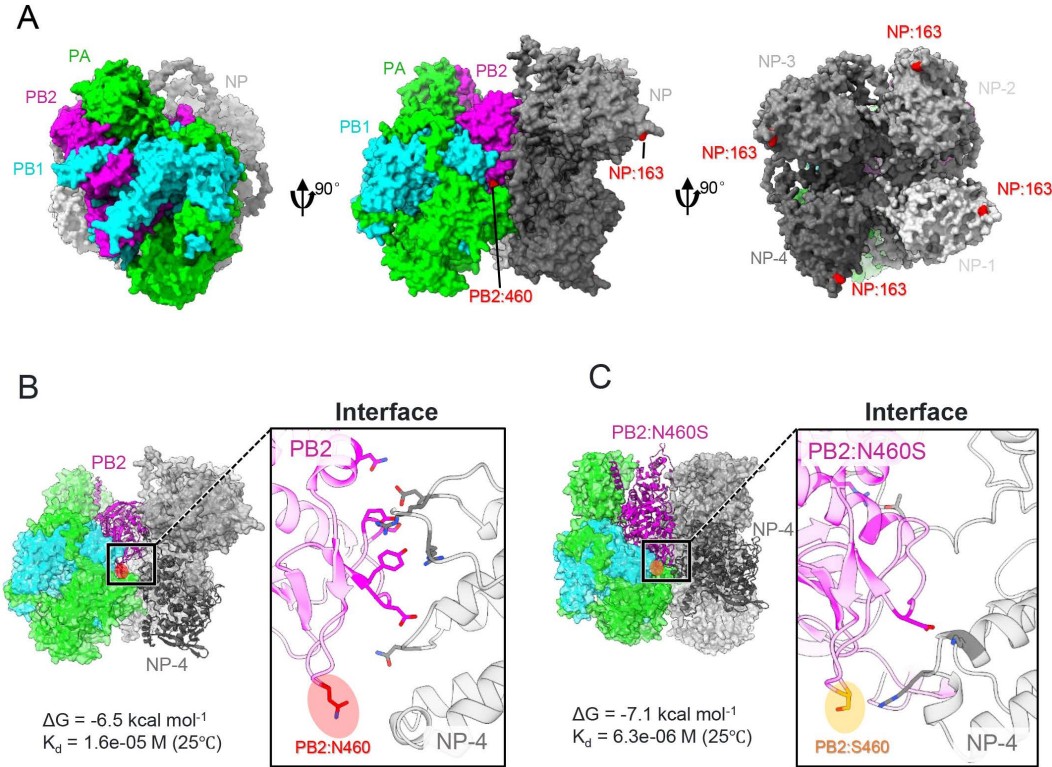

**Fig 6. Structural modeling of the PB2–NP interaction and the effect of the PB2-N460S substitution.** (A) The structure of the GZ0215-01 RNP complex was predicted using AlphaFold 3, and the structure was visualized with ChimeraX 1.7. (B) Polymerase-NP interface derived from GZ0215-01. Left panel: overall complex structure, with PB2 and NP-4 shown as ribbon models. Right panel: zoomed view of the PB2-NP interface, with PB2-460N highlighted in red. (C) PB2-N460S mutant interface. Left panel: overall structure of the mutant polymerase–NP complex. Right panel: zoomed view showing PB2-460S in yellow. Accompanying PRODIGY-calculated binding free energy (ΔG) and dissociation constant (Kd) values are indicated for each complex.

Collectively, these results suggest that the PB2-N460S/NP-I163T substitutions orchestrate a coordinated enhancement of IBV viral replication and pathogenicity by increasing their gene expression, enhancing PB2-NP binding, and improving polymerase activity.

## Discussion

IBVs pose a significant global public health threat, especially when they become the dominant in seasonal circulation [9]. Unlike IAVs, whose pathogenesis is well-studied, the molecular determinants of IBVs pathogenicity remain poorly characterized. Here, we demonstrated that PB2-N460S and NP-I163T substitutions, derived from two viral plaque colonies obtained from a single oropharyngeal swab, synergistically enhance IBV replication and pathogenicity via enhanced protein expression, PB2-NP interaction, and polymerase activity.

Both PB2 and NP are essential components of the RNP complex, which is crucial for the transcription and replication of the viral genome [14]. PB2, a subunit of the polymerase complex, functions as a cap-binding protein, interacting with the cap structure of host mRNAs. This interaction facilitates the endonucleolytic cleavage of host pre-mRNAs, which are then used as primers for synthesizing viral mRNAs through cap-snatching [19]. On the other hand, NP, a major structural protein of the virus, forms multiple copies to package the vRNA or cRNA and interacts with the polymerase, playing a key role in switching between viral RNA replication and transcription [35,37]. Substitutions in both PB2 and NP have been shown to affect viral replication or host tropism [38]. Notable substitutions in IAV PB2 include E627K, D701N [23], T271A [39]. E249G, G309D, T339M [40], K526R [41], I714S [42], while IBV PB2 mutations such as F406Y and W359F have also been identified [31]. For NP, mutations like Q357K [26], N319K [28,29], D88A [27], acetylation positions K77, K113 and K229 [43] have been associated with altered viral growth. Additionally, for IAV, combinations of mutations in PB2 and NP have been reported to synergistically enhance viral growth [44]. For example, the combination of PB2-Q591K and NP-Q12R, S50G, or N309K, as well as PB2-E627K and NP-Q12R, and PB2-Q236H, E627K and NP-N309K [44], underscores the functional relationship between PB2 and NP in viral replication. In the present study, we are the first to identify a paired substitution in both PB2 and NP that affects polymerase activity, viral replication, and pathogenicity, while each individual substitution alone did not have such phenotype (Figs 2–3, and 5). These findings highlight the importance of the combined effect of PB2-N460S and NP-I163T substitutions in enhancing viral fitness, providing a deeper understanding of the molecular mechanisms that govern IBV pathogenesis.

Both PB2-N460S and NP-I163T substitutions elevated protein expression by enhancing mRNA levels (Fig 5). Notably, neither single substitution alone significantly improved polymerase activity, viral replication kinetics, or pathogenicity, indicating a synergistic functional dependency between PB2-N460S and NP-I163T. To investigate potential structural interactions, we employed AlphaFold 3 modeling given the unresolved nature of IBV PB2-NP binding interfaces. The predicted structure localized PB2-460 at the PB2-NP interface—a spatial configuration consistent with IAV PB2 architecture, where residue 460 resides within both the cap-binding domain and NP-interacting region [45]. Conversely, NP-163 was positioned distal to the PB2 interface, aligning with prior IAV studies identifying NP residues 1–161 and 255–498 as critical PB2-binding regions [35]. While these data suggest no direct contact between PB2-460 and NP-163, indirect interaction mechanisms involving RNA or host cofactors remain plausible. This hypothesis is corroborated by coimmunoprecipitation assays showing markedly enhanced PB2-NP coprecipitation efficiency in viruses harboring both substitutions compared to single substitutions.

Phylogenetic analysis of all available influenza B virus sequences retrieved from the GISAID database revealed near-fixation of PB2-460S and NP-163T in annual IBV isolates (>99% prevalence), while the PB2-460N and NP-163I occurs at <4% and <0.2%, respectively. Such overwhelming dominance of PB2-460S and NP-163T strongly implies positive selection, likely through augmenting viral fitness via enhanced protein expression and elevated polymerase activity. The recovery of the rare PB2-460N/NP-163I variant from our patient sample likely reflects genuine intrahost viral

heterogeneity. Our passage-1 whole-genome sequencing on MDCK cells confirmed these substitutions at low frequency, consistent with their lower replication fitness—explaining why they are underrepresented in global prevalence data.

We sequenced viral genomes from both MDCK cell supernatants and mouse lung tissues for all four recombinant viruses (S2 and S3 Tables). Importantly, no genetic polymorphism at the PB2-460 and NP-163 positions were observed across the four viruses, confirming that these genotypes were present in the original clinical sample and did not arise during viral isolation. In MDCK cultures, we observed only low-frequency (< 15%) host-adaptive substitutions at other positions, and in mouse lungs no high-frequency (> 50%) adaptive mutations were detected. A PB2-527 polymorphism appeared at moderate frequency (25–49%) across all variants but was less prevalent in the PB2-460S/NP-163T double mutant than in the other viruses. These findings exclude culture-derived adaptive changes as contributors to the enhanced replication and pathogenicity of the PB2-460S/NP-163T virus. Interestingly, the lower replication variants acquired host-adaptive mutations more readily than the double mutant, suggesting that suboptimal polymerase function may impose stronger selective pressure.

To verify that the PB2-N460S/NP-I163T synergy is not unique to GZ0215, we introduced the reverse mutations (PB2-S460N and NP-T163I) into a contemporary IBV strain, GZ50. In this background, the double-reverse mutant (GZ50-NI) and single mutants (GZ50-NT, GZ50-SI) all exhibited reduced plaque size, slower growth in MDCK cells, and markedly attenuated virulence in mice compared to the parental GZ50-ST (S2 Fig). These findings confirm that restoring PB2-460N/NP-163I attenuates both replication and pathogenicity, demonstrating that the cooperative effect of PB2-N460S/NP-I163T is conserved across diverse IBV lineages.

The PB2-N460S/NP-I163T dual substitutions exhibited enhanced viral replication kinetics and polymerase activity, concomitant with amplified immune activation in the murine challenge model (Figs 3 and 4). Transcriptomic profiling of infected lung tissues demonstrated marked upregulation of pro-inflammatory cytokines (e.g., IL-6, TNF-α) and chemokines (e.g., CCL-5, CCL2), indicative of dysregulated innate immune signaling. Although such inflammatory responses are essential for pathogen containment, excessive activation precipitates immunopathology—a principal mediator of pulmonary epithelial damage and acute respiratory distress syndrome (ARDS) progression [46–49]. Furthermore, our previous study demonstrated that extracellular matrix (ECM) proteases, particularly ADAMTS4, play a key role in the immunopathogenesis of influenza infection by driving excessive lung inflammation and tissue damage [50]. In this study, we observed significantly upregulated secretion of proteases such as Cathepsin S (Ctss) in dual substitution-infected mice. These proteases may contribute to tissue damage, suggesting a convergent protease-mediated mechanism. Such proteolytic activity may exacerbate lung injury via ECM degradation, creating a feedforward loop of tissue damage and inflammatory exacerbation.

In summary, the PB2-N460S and NP-I163T substitutions have been shown to synergistically enhance polymerase activity, viral replication, pathogenicity, and host inflammatory responses in IBVs, providing insight into molecular determinants of IBV pathogenesis and informing future prevention and treatment strategies. These substitutions, when present together, synergistically enhance polymerase activity, promote viral replication, increase pathogenicity, and elevate the host inflammatory response, potentially contributing to the severity of IBV infections. Our findings provide valuable insights into the molecular determinants of IBV pathogenesis and may inform future strategies for improving the prevention and treatment of IBV infections.

## Materials and methods

### Ethics statement

Oropharyngeal swab collection was approved by the Ethics Committee of the First Affiliated Hospital of Guangzhou Medical University (Ethics No. 2016–78). Patient consent was waived due to the anonymized retrospective nature of the residual specimen use. Animal experiments were approved by the Institutional Animal Care and Use Committee of Guangzhou Medical University (Permit No. 20220605) and performed in an Animal Biosafety Level 2 (ABSL-2) facility at the First Affiliated Hospital of Guangzhou Medical University.

## Cells and viruses

Madin-Darby canine kidney cells (MDCK, CCL-34, American Type Culture Collection [ATCC]), human embryonic kidney (HEK) 293H cells (Invitrogen, Thermo Fisher Scientific), and HEK 293T cells (CRL-3216, ATCC) were maintained in Dulbecco's Modified Eagle's Medium (DMEM, Gibco) with 10% fetal bovine serum (Gibco) at 37°C with 5% $CO_2$. The influenza B virus strain B/Guangzhou/0215/2012 (Victoria lineage; referred to as GZ0215) was isolated from a human oropharyngeal swab specimen collected at the First Affiliated Hospital of Guangzhou Medical University, Guangzhou, China. Virus isolation, rescue, propagation, and *in vitro* assays with authentic influenza B virus were carried out under enhanced Biosafety Level 2 (BSL-2+) conditions at the First Affiliated Hospital of Guangzhou Medical University, in compliance with the People's Republic of China Biosafety Law.

## Genomic sequencing and SNP calling

Influenza B virus whole-genome sequencing was performed using the Illumina NovaSeq 6000 platform, as described elsewhere [51]. Paired-end reads obtained from sequencing were aligned to the corresponding Sanger-sequenced whole genome consensus using Qiagen CLC Genomics Workbench (version 22.0.1). A quality threshold of 0.04 and a minimum coverage of 10× were applied for genetic variant analysis. The raw sequencing data collected from this study have been deposited to NCBI Sequence Read Archive under the BioProject accession No. PRJNA1294812.

## Plaque assay

Plaque assays were performed as previously described [52] with minor modifications. Briefly, viruses were serially diluted in phosphate-buffered saline (PBS) and applied to confluent MDCK cells grown in six-well plates. The cells were incubated at 33°C for one hour to allow viral adsorption. After removing the inoculum, the cells were washed twice with PBS to remove excess viruses. Subsequently, the cells were overlaid with a 1:1 mixture of 1.2% low-melting point agarose (Lonza) and 2×Minimum Essential Medium (MEM) supplemented with 0.3% bovine serum albumin (BSA, MP Biomedicals, LLC). The plates were then incubated at 33°C for three days to allow plaque formation.

For plaque morphology determination, plaques were visualized using an immunostaining assay [53]. For viral clone purification, plaques were picked using a sterile 200 µl micropipette tip, and the viruses were collected by puncturing the agarose layer. The collected viruses were then inoculated into 10-day-old SPF embryonated chicken eggs and incubated at 33°C for 72 hours. The viral genomes were confirmed by Sanger sequencing.

## Virus rescue by reverse genetics

The eight genomic segments of GZ0215-01 were amplified using RT-PCR and subsequently cloned into a pM reverse genetics plasmid as described previously [52,54]. For clarity, GZ0215-01 naturally encodes PB2 residues 261M and 460N, PA residues 293N, 338K, and 495M, and NP residue 163I. Site-directed mutagenesis was performed to introduce the specified amino acid substitutions in PB2, and NP into the corresponding reverse genetics plasmids. Virus rescue was carried out as previously described [54]. Briefly, plasmids carrying the eight GZ0215-01-derived gene segments, with or without the introduced mutations, were co-transfected into co-cultured 293H and MDCK cells using Lipofectamine 2000 (Invitrogen). The cells were incubated at 33°C, and 1 ml of Opti-MEM (Gibco) containing 1 µg/ml TPCK-trypsin was added 48 hours post-transfection. The rescued viruses were plaque purified and amplified in 10-day-old specific-pathogen-free (SPF) embryonated chicken eggs. The eight segments of each recombinant virus were confirmed by Sanger sequencing.

## Growth kinetics

To compare the growth kinetics of the indicated viruses, MDCK cells were inoculated with a multiplicity of infection (MOI) of 0.01. After a 1-hour incubation at 33°C, the cells were washed with DPBS and maintained in DMEM containing 0.3%

BSA (MP Biomedicals), 1 µg/ml TPCK-treated trypsin (Sigma Aldrich), and 1% Penicillin-Streptomycin (Invitrogen) at 33°C with 5% $CO_2$. Cell supernatants were collected at 12, 24, 36, 48, 60, and 72 hpi.

Virus titers were determined using an ELISA-based $TCID_{50}$ assay with MDCK cells, as previously described [55]. Briefly, the viral samples were half-log serially diluted on 96-well plates with DMEM medium supplemented with 1 µg/mL of TPCK-trypsin (Sigma Aldrich) and 0.3% BSA (MP Biomedicals). Next, $3 \times 10^4$ MDCK cells were added to each well. Following a 1-day incubation at 33°C with 5% CO2, cells were fixed in 80% acetone, stained with anti-IBV NP rabbit poly-clonal antibody (Cat. No. 40438-T62-100, Sino Biological). This was subsequently detected by the addition of horseradish peroxidase (HRP)-conjugated goat anti-rabbit IgG (Cat. No. E030120-02, EARTH) and TMB-ELISA substrate (Thermo Fisher Scientific). Wells with $OD_{450} > 2 \times$ mean of cell controls were scored positive, and $TCID_{50}$ was calculated by the Reed–Muench method.

## Expression analysis of NP and PB2 at mRNA and protein levels

GZ0215-derived PB2-Myc (with 460N or 460S) and NP (with 163T or 163I) were cloned into the expression vector pCAGGS, as described elsewhere [56]. HEK 293T cells were seeded in 12-well plates and transfected with 1 µg of the indicated plasmid per well (pCAGGS-Myc-PB2 or pCAGGS-NP) using Lipofectamin 2000 Transfection Reagent (Invitro-gen), according to the manufacturer's instruction. At 24 hours post-transfection (hpt), total RNA was extracted from trans-fected cells using the Tissue RNA Purification Kit PLUS (EZBioscience). Plasmid DNA contamination was removed by on-column DNase I treatment (Qiagen). Reverse transcription was performed using PrimeScript RT Master Mix (Takara) with oligo(dT) primers. Quantitative real-time PCR (qRT-PCR) for PB2 and NP genes (S1 Table) was carried out using TB Green Premix Ex Taq II (Takara) in a 7500 real-time PCR system (Applied Biosystems). The hGAPDH gene was used as an internal reference, and relative mRNA expression levels were determined using the $2^{-\Delta\Delta Ct}$ method. In parallel, trans-fected cells were harvested, resuspended in PBS, and lysed with $5 \times$ SDS loading buffer for Western blot analysis. Pro-teins were separated by SDS-PAGE and transferred onto PVDF membranes (Millipore). Protein expression was detected using rabbit anti-NP polyclonal antibody (Cat. No. 40438-T62, Sino Biological) or rabbit anti-Myc tag polyclonal antibody (Cat. No. 16286–1-AP, Proteintech), followed by HRP-conjugated secondary antibodies (Cell Signaling Technology). Sig-nals were visualized using an ECL detection kit (Thermo Fisher Scientific).

## Co-immunoprecipitation (Co-IP) assay

For the PB2 and NP Co-IP assay, HEK 293T cells were seeded in 6 cm dishes and cultured overnight at 37°C. When cell confluence reached >80%, the following four plasmids, carrying the corresponding genes from GZ0215-01 with or with-out substitutions, were co-transfected: pM-PB2-Myc (bearing 460N or 460S), pM-NP (bearing 163T or 163I), pM-PB1, and pM-PA. At 24 hpt, cells were washed with ice-cold PBS and lysed with Pierce IP Lysis Buffer (Thermo Scientific, USA) supplemented with the protease inhibitor 100µM PMSF (Abcam). Cell lysates were centrifuged at $13,000 \times g$ for 30 minutes at 4°C, and the supernatant was collected. The lysates were then incubated overnight at 4°C with Dynabeads magnetic beads (Thermo Scientific) pre-conjugated with anti-NP antibodies (Cat. No. 40438-T62, Sino Biological). The beads were washed three times with PBST, and the bound proteins were analyzed by Western blotting.

## Minigenome assay

The minigenome assay was performed to compare the polymerase activity of RNPs with or without amino acid substitu-tions in PB2 and NP in 293H cells. Briefly, plasmids expressing viral RNP components (PB2, PB1, PA, and NP) derived from the indicated mutants, along with the pPoII-FFLuc-RT plasmid (encoding *firefly* luciferase in the negative-sense orientation) and the pRL-SV40 plasmid (Promega, encoding *Renilla* luciferase as an internal control) were co-transfected

into 293H cells using Lipofectamine 2000 (Invitrogen). After 24 h of incubation, *firefly* and *Renilla* luciferase activities in the transfected cells were measured using the Dual Luciferase Reporter Assay System (Promega), following the manufacturer's instructions. The relative luciferase expression (*firefly*/*Renilla* luciferase ratio) was used to assess the replication efficiency of the RNP complexes. A higher firefly/Renilla ratio indicates higher polymerase activity for the tested RNP set. Each RNP combination was performed in triplicate.

## Mouse experiment

Six-week-old SPF female BALB/c mice (Vital River Laboratories, China) were intranasally inoculated with serial doses of the indicated viruses in a 50 µl volume under anesthesia with isoflurane. The mice were monitored daily for 10–14 days for survival and weight loss following the challenge. Control mice were inoculated with 50 µl of PBS only. Any animal that lost ≥25% of its initial body weight was humanely euthanized. The 50% lethal dose ($LD_{50}$) of each virus was determined using the Reed and Muench method. At 1, 3, and 5 dpi, nasal turbinates and lungs were collected. The left lung lobes were fixed in 10% formalin for histopathological analysis. The right lung lobes and nasal turbinates were homogenized in 1 ml of PBS containing 1% penicillin-streptomycin. Viral titers were determined in MDCK cells by ELISA-based $TCID_{50}$.

## Measurement of cytokine/chemokine expression

Total RNA was extracted from mouse lungs using the Tissue RNA Purification PLUS Kit (EZBioscience). Genomic DNA was removed by on-column DNase I treatment (Qiagen) during the RNA extraction process. The RNA was then reverse transcribed into cDNA using the PrimeScript RT Master Mix (Takara) with random hexamer primers (Thermo Fisher Scientific). The cDNA was used for real-time PCR with TB Green Premix Ex Taq II (Takara) to amplify specific targets (S1 Table). Real-time PCR assays were performed as previously described [57]. Mouse β-actin expression was used as a housekeeping gene to normalize the RNA input from mouse tissues. The $2^{-\Delta\Delta Ct}$ method was used to compare differential gene expression between test samples. The mean fold change ($2^{-\Delta\Delta Ct}$) values of triplicates and their standard deviations are reported.

## Histopathology

Mouse lung tissues were fixed in 10% PBS buffered formalin and embedded in paraffin. The tissue sections cut at 5 µm were affixed on glass slides. Tissue slides were stained with hematoxylin and eosin (H&E) stain and viewed under a light microscope (Nikon).

## RNA sequencing and data analysis

Mice lungs infected with the indicated viruses were harvested at 5 dpi for RNA sequencing. Total RNA was extracted using the Tissue RNA Purification Kit PLUS (EZBioscience). RNA libraries for mRNA sequencing were prepared with the NEBNext Ultra RNA Library Prep Kit for Illumina (NEB), following the manufacturer's instructions. Sequencing was performed on the NovaSeq 6000 platform (Illumina, USA) with 150 bp paired-end reads. Raw sequencing data were processed and trimmed using trim-galore (version 0.6.10) to remove adapter sequences and low-quality nucleotides. High-quality reads were then aligned to the ENSEMBL mouse genome (mm10) transcripts using HISAT2 (version 2.1.0) with default parameters. The mapped reads for RefSeq genes were counted using featureCounts (version 2.0.1). Differential expression genes analysis was conducted after normalization with the DESeq2 R package (version 1.38.3). Gene Ontology (GO) biological process and pathway enrichment analyses of differentially expressed genes were performed using the clusterProfiler package (version 4.6.2) as described previously [58]. The data that support the findings of this study are publicly available from Gene Expression Omnibus with the identifier(s) GSE289500.

## Structural prediction and interaction analysis

The three-dimensional structures of GZ0215-01 RNP complex and its PB2-N460S mutant were predicted *de novo* using AlphaFold3 [59], with five independent runs and selection of the model exhibiting the highest confidence score. Binding free energy (ΔG) and dissociation constant (Kd) between PB2 and NP-4 were calculated using PRODIGY [60,61] under conditions of 25°C. Protein-protein interaction interfaces were systematically analyzed by PDBePISA (https://www.ebi.ac.uk/pdbe/pisa/). Structural visualization and rendering were performed in ChimeraX v1.7 [62].

## Statistical analysis

Statistical analyses were performed using GraphPad Prism version 9. Data are presented as means±SD for all data. For comparisons between two groups, an unpaired t-test was used when variance was equal between groups, while Welch's t-test was applied when variance was unequal. For comparisons involving multiple groups, one-way ANOVA with Tukey's post-hoc test was employed. To achieve normality, viral titer data were log-transformed prior to analysis. *p* value<0.05 was considered statistically significant.

## Supporting information

**S1 Table. Primers used to quantify mRNA expression of IBV genes and mouse-specific proinflammatory markers.**
(DOCX)

**S2 Table. Deduced amino acid substitutions in influenza B viruses harvested from MDCK cells at 72 hpi following challenge with the indicated strains.**
(XLSX)

**S3 Table. Deduced amino acid substitutions in influenza B viruses collected from mouse lung tissues at 5 dpi following challenge with the indicated strains.**
(XLSX)

**S1 Fig. (related to** Fig 3**).** Body weight changes (left panels) and survival rates ((right panels) of six-week-old female BALB/c mice intranasally infected with $10^2$ (A), $10^3$ (B), $10^5$ (C)TCID$_{50}$ of indicated recombinant IBVs.
(TIF)

**S2 Fig. PB2-N460S and NP-I163T substitutions do not affect viral attachment or internalization in MDCK cells (related to** Fig 2**).** (A) NP RNA of viruses attached to the surface of MDCK cells. MDCK cells were incubated with each recombinant virus at 4 °C for 1 h to permit attachment, washed with ice-cold PBS (pH 7.2) to remove unbound virus, and cell-attached NP RNA was quantified by RT-qPCR. (B) NP RNA of viruses internalized into MDCK cells. MDCK cells were incubated with each recombinant virus at 4 °C for 1 h, transferred to 33 °C for 1 h to allow internalization, then washed with acidic PBS (pH 1.5) to remove remaining surface-attached virus. Intracellular NP RNA was measured by RT-qPCR. ns, not significant.
(TIF)

**S3 Fig. Functional validation of PB2-N460S/NP-I163T synergy in the GZ50 background.** Reverse mutations (PB2-S460N and NP-T163I) were introduced into a more recently circulating IBV strain, B/Guangzhou/50/2022 B/Guangzhou/50/2022 (GZ50; Victoria lineage, clade V1A.3a.1; GISAID isolate EPI_ISL_19888228) by using reverse genetics. Four viruses were generated on the GZ50 backbone: GZ50-NI (PB2-460N/NP-163I; double-reverse mutant), GZ50-SI (PB2-460S/NP-163I; single-substitution mutant), GZ50-NT (PB2-460N/NP-163T; single-substitution mutant), GZ50-ST (parental PB2-460S/NP-163T). (A) Representative plaque assays of these four GZ50-derived viruses on MDCK cells, stained 72 h post-infection. (B) Plaque diameters for each recombinant IBV were determined by Adobe Photoshop (CC

2019). (C) Growth kinetics of the indicated recombinant viruses in MDCK cells at an MOI of 0.01. Statistical significance is indicated using asterisks in different colors for clarity: black, blue, and green asterisks denote significant differences of GZ50-NI vs. GZ50-ST, GZ50-SI vs. GZ50-ST, and GZ50-NT vs. GZ50-ST, respectively. Body weight changes (D) and survival rates (E) of six-week-old female BALB/c mice (n = 4 per group) intranasally infected with $10^4$ $TCID_{50}$ of each GZ50-derived virus or PBS. (F) Lung viral titers measured at 1, 3, and 5 dpi (n = 3 per time point) following infection with $10^4$ $TCID_{50}$ of the indicated viruses. The limit of detection is indicated by the horizontal dashed line. * $p < 0.05$; **, $p < 0.01$; ***, $p < 0.001$; ****, $p < 0.0001$; ns, not significant.
(TIF)

**S4 Fig. Correlation of ssGSEA-derived innate immune response scores with disease phenotypes in mice (related to Fig 4).** Single-sample gene set enrichment analysis (ssGSEA) enrichment scores for the "Regulation of innate immune response" pathway were calculated from lung transcriptomes of mice infected with rgPB2$_{460N}$/NP$_{163I}$ (NI) or rgPB2$_{460S}$/NP$_{163T}$ (ST) viruses at 5 dpi. Panels show the correlation between ssGSEA scores and body-weight loss (A), viral titer in nasal turbinate (B), and viral titer in lung tissue at the same timepoint (C). Regression lines (solid) and 95% confidence intervals (shaded) are displayed. Pearson's R and $p$ values are indicated in red.
(TIF)

## Author contributions

**Conceptualization:** Weiqi Pan.

**Data curation:** Yang Wang, Yuting Ye, Chunguang Yang.

**Formal analysis:** Yang Wang, Tianxin Ma, Yuting Ye.

**Funding acquisition:** Zhiqi Zeng, Zifeng Yang, Weiqi Pan.

**Investigation:** Yang Wang, Yu Gao, Tianxin Ma, Chenyang Cao, Binqian Zou, Sulan Ye, Qingsheng Huang, Shengfeng Li, Lixi Liang, Hongxuan Zhou, Zhiqi Zeng.

**Methodology:** Yang Wang, Yu Gao, Tianxin Ma.

**Project administration:** Yang Wang.

**Supervision:** Yang Wang, Zifeng Yang, Weiqi Pan.

**Validation:** Zifeng Yang.

**Visualization:** Yang Wang, Yu Gao, Tianxin Ma, Yuting Ye, Chenyang Cao, Binqian Zou.

**Writing – original draft:** Yang Wang, Yu Gao.

**Writing – review & editing:** Yang Wang, Chenyang Cao, Zifeng Yang, Weiqi Pan.

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
