## [Decision Letter · Decision Letter 0]

24 Apr 2025

PPATHOGENS-D-25-00454

N460S in PB2 and I163T in nucleoprotein synergistically enhance the viral replication and pathogenicity of influenza B virus

PLOS Pathogens

Dear Dr. Wang,

Thank you for submitting your manuscript to PLOS Pathogens. After careful consideration, we feel that it has merit but does not fully meet PLOS Pathogens's publication criteria as it currently stands. Therefore, we invite you to submit a revised version of the manuscript that addresses the points raised during the review process.

Please submit your revised manuscript within 60 days Jun 23 2025 11:59PM. If you will need more time than this to complete your revisions, please reply to this message or contact the journal office at plospathogens@plos.org. Please include the following items when submitting your revised manuscript:

We look forward to receiving your revised manuscript.

Kind regards,

Linda Brunotte

Guest Editor

PLOS Pathogens

Thomas Hoenen

Section Editor

PLOS Pathogens

 Sumita Bhaduri-McIntosh

Editor-in-Chief

PLOS Pathogens

orcid.org/0000-0003-2946-9497

 Michael Malim

Editor-in-Chief

PLOS Pathogens

orcid.org/0000-0002-7699-2064

**Additional Editor Comments:**

PLOS and PLOS Pathogens have a specific policy regarding research that would fall into the category of “dual use research of concern” (https://journals.plos.org/plospathogens/s/ethical-publishing-practice#loc-biosecurity-and-dual-use-research-of-concern). Your submission was referred to the PLOS Dual Use Committee for an assessment of the potential risks versus benefits of publication. The Committee determined that there are some potential risks of publishing your submission, but these may be mitigated if the authors can address the following concerns. Please address these as part of your revisions, and include your responses in your Response to Reviewers when you resubmit. Your responses may be referred back to the DURC Committee to confirm that the risks have been adequately mitigated, so please ensure your revisions are thorough.

The authors should include details about biosafety and containment measures as well as institutional and regulatory oversight of the work conducted.The authors should include some more information as to the rationale for this study, and provide additional details of the clinical context that may be relevant to the evaluation of potential biosecurity concerns.Why were these isolates specifically chosen for further experimentation?Were there specific clinical symptoms from the patient that prompted further investigation?Was the initial culture of the isolates routine, and the increased plaque size of one of the isolates was the prompt for further investigation?What was the sequence of experiments for this study?
The first line of the concluding statement could be misinterpreted as recommending the best way to enhance the pathogenicity of IBV. This sentence should be rephrased in a more passive voice to prevent any assumptions of DURC or misinterpretation.

**Journal Requirements:**

At this stage, the following Authors/Authors require contributions: Yu Gao, Tianxin Ma, Yuting Ye, Chunguang Yang, Lixi Liang, Hongxuan Zhou, Sulan Ye, Qingsheng Huang, Shengfeng Li, Zhiqi Zeng, Yang Wang, and Weiqi Pan. Please ensure that the full contributions of each author are acknowledged in the "Add/Edit/Remove Authors" section of our submission form.

- ® on page: 11

- TM on pages: 10, and 11.

4) We notice that your supplementary Table is included in the manuscript file. Please remove it and upload it with the file type 'Supporting Information'. Please ensure that each Supporting Information file has a legend listed in the manuscript after the references list.

5) Thank you for stating “The data that support the findings of this study are publicly available from Gene Expression Omnibus with the identifier(s) GSE289500.” Please note that, though access restrictions are acceptable now, your entire minimal dataset will need to be made freely accessible if your manuscript is accepted for publication. This policy applies to all data except where public deposition would breach compliance with the protocol approved by your research ethics board. If you are unable to adhere to our open data policy, please kindly revise your statement to explain your reasoning and we will seek the editor's input on an exemption.

1) Please clarify all sources of financial support for your study. List the grants, grant numbers, and organizations that funded your study, including funding received from your institution. Please note that suppliers of material support, including research materials, should be recognized in the Acknowledgements section rather than in the Financial Disclosure

2) State the initials, alongside each funding source, of each author to receive each grant. For example: "This work was supported by the National Institutes of Health (####### to AM; ###### to CJ) and the National Science Foundation (###### to AM)."

3) State what role the funders took in the study. If the funders had no role in your study, please state: "The funders had no role in study design, data collection and analysis, decision to publish, or preparation of the manuscript."

4) If any authors received a salary from any of your funders, please state which authors and which funders.

7) Your current Financial Disclosure states, "The author(s) received no specific funding for this work."

However, your funding information on the submission form indicates receiving funds. Please ensure that the funders and grant numbers match between the Financial Disclosure field and the Funding Information tab in your submission form. Note that the funders must be provided in the same order in both places as well.

**Reviewers' Comments:**

Reviewer's Responses to Questions

**Part I - Summary**

Reviewer #1: This study identifies the critical roles of PB2-N460S and NP-I163T mutations in enhancing IBV replication and pathogenicity. The co-existence of these substitutions synergistically enhances polymerase activity, promotes viral replication, increases pathogenicity, and amplifies host inflammatory responses, potentially contributing to severe IBV infections. While demonstrating novelty, the manuscript requires substantial improvements in result interpretation, logical coherence, and experimental validation. Key concerns include insufficient discussion depth, ambiguous conclusions, and methodological uncertainties that necessitate comprehensive revisions.

Reviewer #2: The findings highlight the critical role of the PB2-N460S and NP-I163T substitutions in enhancing the replication and pathogenicity of IBVs. These substitutions, when present together, synergistically enhance polymerase activity, promote viral replication, increase pathogenicity, and elevate the host inflammatory response, potentially contributing to the severity of IBV infections.These findings provide new insights into the molecular determinants of IBV pathogenesis, highlighting the synergistic effect of PB2-N460S and NP-I163T in enhancing viral fitness and worsening disease outcomes.However, there are several areas that require further clarification and improvement.

Reviewer #3: The authors identified two amino acid substitutions—N460S in PB2 and I163T in NP—from IBV clinical isolates with distinct replication and pathogenicity phenotypes. Using reverse genetics, they demonstrated that while neither mutation alone significantly impacted viral replication or pathogenicity, their combination synergistically enhanced both traits. These findings offer valuable insights into the functional interplay between PB2 and NP, key components of the viral RNP complex. However, the approach lacks novelty, and the extent to which the results can be generalized to other IBV strains remains unclear, limiting the overall impact of the study.

Reviewer #4: In the study entitled “N460S in PB2 and I163T in nucleoprotein synergistically enhance the viral replication and pathogenicity of influenza B virus” by Gao et al., the authors assess molecular determinants of Influenza B virus (IBV) pathogenesis. To this end, genomic analysis of two influenza B virus (IBV) isolates, recovered from a patient’s single swab specimen, was performed and identified two non-synonymous substitutions – PB2-N460S and NP-I163T – as potential molecular drivers of phenotypic differences. Recombinant IB viruses carrying either or both mutations were generated and utilized in in vitro and in vivo approaches to demonstrate that individual mutations had no significant impact while their combination enhanced polymerase activity, replication efficiency, and pathogenicity, with severe disease outcome in the mice animal model. Next, Gao et al. study the effect of PB2-N460S and NP-I163T substitution on host immune response revealing an enhanced immune activation with elevated expression of antiviral and pro-inflammatory genes in transcriptomic profiles. Finally, using co-immunoprecipitation approaches and minigenome reporter assay, the authors demonstrate an enhanced PB2-NP binding and higher polymerase activity in the combinatorial presence of 460S and 163T. Structural modeling positioned PB2-N460S at the PB2-NP interface and NP-I163T at a distal site, suggesting indirect functional coupling.

This is an interesting study with valuable information on molecular determinants of Influenza B viruses. Although the study design is mostly sound, there are a few comments that the authors should consider to improve the quality and clarity of this manuscript.

**Part III – Minor Issues: Editorial and Data Presentation Modifications**

Reviewer #1: (No Response)

Reviewer #2: (No Response)

Reviewer #3: (No Response)

Reviewer #4: Comments:

1. Please provide descriptions of the disease severity and disease outcome of the IBV patient from whom the isolate B/Guangzhou/0215/2012 (GZ0215) originated. Additionally, was the patient under treatment? This information is relevant for evaluating the clinical relevance since the manuscript discusses pathogenesis.

2. Please provide more details on how patients were selected for inclusion or exclusion into the study to ensure the selection criteria was unbiased.

3. Ethic statement (line 313): “…retrospective nature of the residual sample collection”. Since amino acid substitutions may arise under selective pressure during antiviral therapy please clarify whether the sample used in this study was the initial diagnostic sample (diagnostic leftover) collected on admission to the hospital, or a follow-up sample.

4. Please clarify which material was used to study polymorphism – was the RNA retrieved directly from the oropharyngeal swab, or from the cell culture isolate prior to plaque purification? This distinction is important because if whole genome sequencing was performed on RNA derived from cell culture isolates rather than directly from the clinical sample, non-synonymous substitutions may have been introduced during virus propagation. That could influence the interpretation of polymorphism data and may not accurately reflect the viral population present in the patient.

5. Ethics statement: Please indicate whether mice died during your mouse experiments or were euthanized as endpoint criteria were met at the respective time points. If endpoint criteria apply, please provide.

6. Please clarify which infection doses were used for GZ0215-01 and GZ0215-06 as there appears to be inconsistencies between the results section (line 114) and infection doses presented in Figure 1 panel C and D. Why were different infection doses used?

7. An interesting finding is the increased mRNA expression levels of the PB2-460N and NP-163T variants compared to the respective constructs without amino acid substitution. The coding regions of PB2 and NP were cloned into the pCAGGS vector under the control of an RNA polymerase II promoter, and expression experiments were performed in the absence of other viral components potentially affecting expression. Please explain how this enhanced mRNA level could be achieved despite only a single nucleotide difference between the wild-type and mutant constructs.

Additionally, were these results based on biological replicates or technical replicates within a single experiment?

8. Clarification is needed on the mutations that were introduced into the PA and which PA variant was used throughout your experiments. For instance, in the section “Virus rescue by reverse genetics” (line 349-351), which amino acid substitutions were introduced into the PA. In line 352 and 398-399, please clarify whether the mutations refer only to NP and PB2, or to PA as well.

9. Discussion (line 282-283): The sentence refers to a high prevalence of PB2-460S and NP-163T substitutions in IBV sequences retrieved from the GISAID database. However, the term “high” remains vague. Please provide exact proportions of sequences carrying these substitutions and indicate whether PB2-460S and NP-163T independently or in combination or both is meant in the statement. What about the spatial and temporal distribution of sequences with PB2-460S and NP-163T (have thy been consistently present in circulating strains over an extended time period or is there a recent increase).

10. It appears that a previous publication reporting on Influenza B virus animal models [https://doi.org/10.1186/s12985-019-1171-3] used the same patient sample employed in the current study. To aid the reader’s understanding, I recommend citing the earlier study and briefly contextualize the results.

11. Results (line 123–124): “we employed reverse genetics to generate four recombinant IBV viruses in the GZ0215-01 genetic background using reverse genetics”. The term “reverse genetics” is duplicated and should be removed.

12. Results (line 212–213): “It has been reported that the role of PB2-NP interactions in viral ribonucleoprotein (vRNP) assembly and RNA synthesis regulation [25, 26].” The statement of the sentence is unclear since essential parts are missing. Please rephrase for clarity.

13. Histopathology (line 433): Please verify the statement that “5 mm” tissue sections were mounted on glass slides – this appears unusually thick for histopathology.

14. Table 1 (line 655): There appears to be a mistake at amino acid positions 460 in PB2. The codons AGT (Serine) and AAT (Asparagine) do not match the order of the indicated residues. Please verify and correct.

15. The caption of Figure 1 seems to be mislabeled and should be revised. Specifically, (line 667): “(C) Body weight changes and (D) survival rates” does not match the data shown. Additionally, line 669 – 671 “(E) The median lethal doses (LD50) of the two isolates were calculated based on the survival rates from panel D. Whole viral genome analysis via Sanger sequencing…” should be clarified.

16. Caption of Figure 1 (line 672): “resulted from codon changes aat1401-1403agt and atc547-549acc, respectively” The nucleotide position stated here differs from what is shown in Figure 1 panel (E) and Table 1. Please verify.

17. Figure 4 (line 710): There is no panel (E) presented in the figure referenced. Please revise.

18. Formatting: Please check the format of the p values and references. Reference 9 includes a partial “competing interest” statement and should be revised, while few references do not provide a digital object identifier (doi).

PLOS authors have the option to publish the peer review history of their article (what does this mean? ). If published, this will include your full peer review and any attached files.

**Do you want your identity to be public for this peer review?** For information about this choice, including consent withdrawal, please see our Privacy Policy .

Reviewer #1: No

Reviewer #2: No

Reviewer #3: No

Reviewer #4: No

**Part II – Major Issues: Key Experiments Required for Acceptance**

Reviewer #1: 1. paragraph 77-78: The statement regarding NP protein distinctions between IAVS and IBVS requires expansion. Please provide explicit comparative analysis of their structural/functional differences rather than simply stating dissimilarity.

2. paragraph 115-117: Why do GZ0215-01 and GZ0215-06 have different pathogenicity in mice with the same infection dose? If GZ0215-01 has low replication ability, the same infection dose should contain more virus particles. Is it because of the difference in the ability of MDCK cells and mouse cells to replicate the virus? The authors should explore this issue and provide their TCID50 detection method and results in the supporting documents.

3. paragraph 149-154: Potential adaptive mutations during in vivo passage must be systematically excluded. Include sequencing data from post-infection isolates to confirm phenotypic stability.

4. paragraph 169-170:The causality between enhanced inflammation and mutations requires rigorous validation. The observed immune responses may represent secondary effects of increased viral load rather than mutation-specific immunomodulation. Revise conclusions accordingly and address this confounding factor in transcriptome analysis.

5.paragraph 206-207: The mRNA upregulation level is greater than the protein expression upregulation level. Does this indicate that 163T is at a disadvantage in the expression stage?

6. paragraph 229-233: Please run interface analysis and affinity analysis to quantify whether and how the characterization is enhanced.

7. paragraph 241: The author repeatedly mentioned the pathogenic mechanism, but the main cause of the pathogenicity mentioned in the article is the enhanced viral replication, which is the reason for the degree of pathogenicity, not its pathogenic mechanism. Body weight changes were also found in mice challenged with the original strain.

8. paragraph 254: Please cite references

9. paragraph 282-287: Regarding PB2-460S and NP-163T, the prevalence should not only be mentioned in the discussion, but should be more intuitively shown in the results, especially the currently prevalent V1A.3a.2 paragraphage viruses.

10.paragraph 304-309: In addition to the current prevalent situation, the authors should also focus on whether GZ0215-06 can be used to replace the high-yield vaccine backbone or whether GZ0215-01 can be used in the design of attenuated vaccines.

Reviewer #2: 1.The authors mention in lines 253-254 of the Discussion that amino acid variations in PB2 and NP may affect viral replication and host tropism. While this study primarily demonstrates that the combined mutations enhance viral replication and pathogenicity in cellular and mouse models, could these mutations also alter cross-species infectivity?

2.Line 277: There is a missing space between "between" and "PB2-460"

3.Figure 2C shows that the combined mutant strain exhibits higher viral load. Could you provide the viral entry load at 2 hours post-infection? This would help determine whether there are differences in viral entry efficiency into cells.

3. For Figure 5D, statistical analysis is missing for the right panel.

4. The research showed that the PB2 - N460S/NP - I163T dual substitutions led to enhanced immune activation and excessive inflammation. How do you think this hyper - inflammatory state relates to the development of drug resistance in influenza B virus? Are there any potential implications for antiviral therapy?

5. Does the dual mutation lead to a higher rate of severe cases clinically compared to the control group?

6. In Figure 5D, the expression of 163I-NP appears notably low despite high loading control levels. Since the authors used plasmid transfection for protein expression, could this result from low transfection efficiency? Including an HA-tagged plasmid as a transfection efficiency control would help clarify this.

7.Figures 3D and 3E show that the combined mutations result in higher viral loads in nasal turbinates and lungs. Could the author further analyze the differences in viral loads between upper and lower respiratory tracts? Would such differences correlate with disease severity?

Reviewer #3: 1. It would greatly enhance the manuscript if the authors evaluated the stability of the reverse-genetic viruses after 72 hours of replication in MDCK cells. Specifically, the emergence of any additional mutations should be investigated. Likewise, it would be valuable to determine whether further mutations arise after 5 days post-infection (dpi) in mice.

2. The authors have not assessed the frequency of the PB2-N460S and NP-I163T mutations in circulating seasonal influenza B virus (IBV) strains.

3. It would be interesting to investigate whether the PB2-N460S and NP-I163T mutations synergistically enhance viral replication and pathogenicity when present in different influenza B virus backgrounds.

**Figure resubmission:**
---

## [Decision Letter · Decision Letter 1]

14 Aug 2025

Dear Dr. Wang,

We are pleased to inform you that your manuscript 'N460S in PB2 and I163T in nucleoprotein synergistically enhance the viral replication and pathogenicity of influenza B virus' has been provisionally accepted for publication in PLOS Pathogens.

Best regards,

Linda Brunotte

Guest Editor

PLOS Pathogens

Thomas Hoenen

Section Editor

PLOS Pathogens

Sumita Bhaduri-McIntosh

Editor-in-Chief

PLOS Pathogens

orcid.org/0000-0003-2946-9497

Michael Malim

Editor-in-Chief

PLOS Pathogens

orcid.org/0000-0002-7699-2064

Reviewer Comments (if any, and for reference):

Reviewer's Responses to Questions

**Part I - Summary**

Reviewer #1: I have carefully reviewed the revised version of the manuscript. The authors have addressed most of the comments raised during the first round of review, and the quality of the work has improved substantially. The revisions have clarified the study’s methodology and strengthened the overall presentation.

However, I recommend further improvement in the Discussion section. While the content is generally sound, certain sentences could benefit from refinement in logical flow and scientific rigor. Enhancing the coherence of arguments and ensuring that interpretations are well-supported by the data would further strengthen the manuscript.

Reviewer #2: The author supplemented experiments or added discussion explanations to address the concerns and questions of the reviewers. So I suggest the editor consider accepting this manuscript, as the impact of point mutations on influenza virus infection and pathogenicity should indeed be given attention.

Reviewer #4: (No Response)

**Part II – Major Issues: Key Experiments Required for Acceptance**

Reviewer #1: (No Response)

Reviewer #2: (No Response)

Reviewer #4: (No Response)

**Part III – Minor Issues: Editorial and Data Presentation Modifications**

Reviewer #1: (No Response)

Reviewer #2: (No Response)

Reviewer #4: (No Response)

PLOS authors have the option to publish the peer review history of their article (what does this mean? ). If published, this will include your full peer review and any attached files.

**Do you want your identity to be public for this peer review?** For information about this choice, including consent withdrawal, please see our Privacy Policy .

Reviewer #1: No

Reviewer #2: **Yes: ** Weijin Huang

Reviewer #4: No

---

## [Editor Report · Acceptance letter]

Dear Dr. Wang,

We are delighted to inform you that your manuscript, "N460S in PB2 and I163T in nucleoprotein synergistically enhance the viral replication and pathogenicity of influenza B virus," has been formally accepted for publication in PLOS Pathogens.

Best regards,

Sumita Bhaduri-McIntosh

Editor-in-Chief

PLOS Pathogens

orcid.org/0000-0003-2946-9497

Michael Malim

Editor-in-Chief

PLOS Pathogens

orcid.org/0000-0002-7699-2064